# Miniature Optical Particle Counter and Analyzer Involving a Fluidic-Optronic CMOS Chip Coupled with a Millimeter-Sized Glass Optical System [note 1]

**DOI:** 10.3390/s21093181

**Published:** 2021-05-03

**Authors:** Gabriel Jobert, Pierre Barritault, Maryse Fournier, Cyrielle Monpeurt, Salim Boutami, Cécile Jamois, Pietro Bernasconi, Andrea Lovera, Daniele Braga, Christian Seassal

**Affiliations:** 1CEA-LETI Minatec, Université Grenoble-Alpes, F-38000 Grenoble, France; pierre.barritault@cea.fr (P.B.); maryse.fournier@cea.fr (M.F.); cyrielle.monpeurt@orange.fr (C.M.); sboutami@hotmail.com (S.B.); 2Institut des Nanotechnologies de Lyon, UMR CNRS 5270 Ecole Centrale de Lyon, Université de Lyon, F-69134 Ecully, France; cecile.jamois@insa-lyon.fr (C.J.); christian.seassal@ec-lyon.fr (C.S.); 3FEMTOprint, CH-6933 Muzzano, Switzerland; pietro.bernasconi@femtoprint.ch (P.B.); andrea.lovera@femtoprint.ch (A.L.); daniele.braga@fluxim.com (D.B.)

**Keywords:** air quality, particulate matter sensor, light-scattering, CMOS image sensor, miniature optics, micro-fabrication

## Abstract

Our latest advances in the field of miniaturized optical PM sensors are presented. This sensor combines a hybrid fluidic-optronic CMOS (holed retina) that is able to record a specific irradiance pattern scattered by an illuminated particle (scattering signature), while enabling the circulation of particles toward the sensing area. The holed retina is optically coupled with a monolithic, millimeter-sized, refracto-reflective optical system. The latter notably performs an optical pre-processing of signatures, with a very wide field of view of scattering angles. This improves the sensitivity of the sensors, and simplifies image processing. We report the precise design methodology for such a sensor, as well as its fabrication and characterization using calibrated polystyrene beads. Finally, we discuss its ability to characterize particles and its potential for further miniaturization and integration.

## 1. Introduction

### 1.1. Particulate Matter and Air Quality Monitoring

Among air pollutants, there is a class defined as Particulate Matter (PM), which consists of a set of particles suspended in the air. The size of these particles can range from a few nanometers up to a few tens of micrometers. PM comes in a wide variety of shapes and chemical compositions, originating from natural or anthropogenic processes. Anthropogenic PM poses serious sanitary issues [1,2]; for instance, it can cause cardiac, cardiovascular, respiratory and neuro-degenerative diseases.

In regulatory texts, PM is ranked into subclasses according to granulometry. Here we focus on the PM_2.5_ subclass (particles smaller than 2.5 µm) that is a known carcinogenic agent (group 1, IARC classification) [3,4]. It has been shown that exposure to PM_2.5_ can cause premature death [5]. Additionally, the PM_1_ subclass (smaller than 1 µm), which is unregulated at the moment, can cause more harm, due to its capacity to penetrate deeply into the human respiratory system [6,7,8].

Beyond health considerations, anthropogenic PM pollution appears to have an adverse effect on climate [9,10], especially for Black-Carbon (BC) soot. Indeed, meteorological models suggest that controlling the emissions of BC could be one of the most efficient ways to mitigate global warming, even more than controlling carbon monoxide or methane emissions [11,12,13,14].

Air quality monitoring (PM, gaseous pollutants, relative humidity, etc.) is classically carried out by stations with laboratory-grade instruments [15]. The major disadvantage of such a monitoring model is that it does not allow the collection of pollution levels with sufficient spatial resolution (e.g., street-to-street air quality, or indoor air quality [16,17,18]). With the emergence of connected on-board systems (IoT, Internet of Things) equipped with geo-localization modules, we see the possibility that a network of portable environmental sensors can measure pollution levels with high spatio-temporal resolution [19]. Such monitoring networks could also achieve great accuracy by sharing and correcting measurements with data from weather conditions (humidity, temperature) [20] and measurements provided by climate monitoring satellites [21,22,23,24,25].

However, an air quality monitoring model based on a broad sensor network is only possible with widely distributed sensors. This underlines the need for inexpensive and portable air quality monitors [26,27].

### 1.2. Toward Further Miniaturization of Optical PM Sensors

Many techniques can be used to measure PM_2.5_ concentrations [28]; however, the optical method is preferred for inexpensive and portable PM_2.5_ sensors thanks to their simplicity and good sensitivity, especially light-scattering sensors [29,30]. The general setup involves a light source that illuminates a sensing volume where particles can scatter light. An off-axis photodetector then captures this scattered light, while avoiding the illumination beam [31].

Numerous research groups are actively working on down-scaling optical light-scattering PM_2.5_ & PM_1_ sensors. The methodology usually involves silicon microfabrication techniques and co-integration of optical and electro-optical elements and air micro-fluidics, the end goal being the full integration of a PM sensor ‘on-a-chip’ [32,33,34,35].

A simple light-scattering setup is able to perform an indirect measurement of the total mass concentration of the sampled aerosol. However, its composition and size distribution is unknown. An inertial element can filter PM_2.5_ from PM_10_ before the optical sensing stage [36]. For instance, this can be achieved by fluidic devices like cyclones [37] or Virtual Impactors (VI) [38].

For a particle of given size, morphology and optical index, we can describe the theoretical angular distribution of the scattered light (what we call the ‘scattering signature’); for example, using the Lorenz–Mie theory applied to spherical particles, which will be discussed in the next subsection. Retrieving an experimental signature enables the estimation of optical and geometrical parameters of particles (and subsequent classification) by performing an inverse problem. For example, precise scattering angular photometry can be achieved by using a rotating detection arm or periscope [39,40] with an optical Fourier transform imaging setup that uses an array of photodetectors coupled with mirrors [41] or lenses [42].

Optical Particle Counters (OPC) can usually be described as a light-scattering sensor [43] that can detect only one particle at once. The use of focusing beam optics and/or particle focusing nozzles is an efficient way to restrict the sampling volume [44].

In this document, we report the development of an OPC that records a particle’s angular scattering efficiency (scattering signature) by using a Fourier imaging setup. This setup is designed to enable the retrieval of the particle’s diameter and refractive index [45] for further classification, as explained earlier.

### 1.3. Lorenz-Mie Theory and Scattering Signatures

Modelling and understanding the phenomenon of elastic light scattering by particles is of high importance and enables the design of analysis systems capable of classifying particles (particle number, size distribution, material, geometry etc.) in order to, for example, characterize ambient air and identify environmental or sanitary hazards, or to create alarm systems that are less sensitive to false positives, in order to operate in contaminated environments.

There are many models that take into account several parameters such as lighting conditions, particle morphology, orientation, size range, multiplicity of particles, etc. Here, we will mainly focus on the Lorenz–Mie theory [46,47]. This is an electromagnetic theory valid for a homogeneous sphere with an arbitrary diameter that is illuminated by a monochromatic plane wave. It shows, notably, that larger particles (when compared to the illumination wavelength) exhibit oscillations in the scattering signature which are not seen for smaller particles. The width and spacing of those oscillations is related to both the sphere’s diameter and refractive index.

Other, broader theories can be applied to model scattering by non-spherical particles or aggregates, as reported in references [48,49].

## 2. Designing the Optical System

### 2.1. Holed Retina, an Opto-Fluidic CMOS Chip

In order to properly record particle scattering signatures, a CMOS (or ’Complementary metal oxide semi-conductor’, which refers to the manufacturing technology) imager chip (referred to as the ‘holed retina’) has been developed and customized for our specific requirements [50] on the basis of the CreaPYX standard platform (by Pyxalis) [51]. The specificity of our design is to allow the formation of a fluidic channel through the chip. The latter were manufactured first through a 200 mm CMOS shuttle run (multi-project wafer) of an undisclosed industrial fab (see, Figure 1a). The fabrication is followed by a specifically developed deep etching post-process performed in-house (200 mm MEMS foundry). This allowed us to drill vertical fluidic channels at wafer scale. The chips are then diced and wire-bonded (ball bonding) onto a ceramic carrier that is also drilled (see, Figure 1b).

The overall footprint of the system is that of the ceramic package (29.2 mm, square). Note that the chip is only sized 6 mm × 5.9 mm. In the future, the CMOS chip will be bonded directly onto PCB, in order to further reduce the overall footprint.

In Figure 2a, we present an optical image of the manufactured holed retina. It is composed of an uncommon dual matrix core part that features two detection areas (sub-retina 1 and 2) spaced by an empty area where a fluidic channel is drilled (oblong profile, 2 mm × 0.5 mm). The size of both matrices is by 75 × 281 pixels, with a 10 µm pixel’s pitch.

Image examples are shown in Figure 2b. Lines 1–75 and 76–150 correspond to the first and second sub-retinae, respectively. In the image, we see a discontinuous image with two separate fields that have been stitched together, caused by the blind central part. The pixel has a 5T architecture, and allows for a global-shutter low-noise readout mode, with two gains high dynamic range [52]. More details on the holed retina concerning the design, fabrication and characterization are provided in reference [31].

### 2.2. Difficulties Encoutered Using a Lens-less Setup

In reference [31], we have used the holed retina in a first generation of optical PM sensor, which is the base frame for the second generation presented here. The former is able to count particles and record scattering signatures that are projected onto the retina using a lens-less configuration. With this first-generation PM sensor, the Field of View (FoV) is limited by the geometry as we can evaluate scattering signatures within only a few tens of degrees. There is little margin of improvement that can be made using lens-free planar projection. Moreover, the uncertain position of the particle within the illumination beam can result in an affine transformation of the image as well as a blurring effect, with a shifted FoV. This last part was taken into account by our image analysis procedure but one may want to simplify the computing component as much as possible for future portable applications.

We address those identified problems through the design and fabrication of a miniature glass optical system, which will be described in the following.

### 2.3. Achieving Position Insensitivity

In most non-miniaturized light-scattering photometers, the position of the particle has little impact on the measurements. Indeed, the range of positions accessible by the particle is very small as compared to the optical path of scattered rays. In our case, the impacts of the particle displacement include translation, magnification and rotation of the recorded image, as it was seen in our reference [31]; a superposition of shifted signatures can be generated by a distribution of particles, resulting in a position blur; as a corollary, the detection of a moving particle (with respect to the image integration time) results in a motion blur.

We will propose a way to achieve position insensitivity. A setup equivalent to a far-field setup is the common d-f lens system (or Fourier-domain imaging setup), where the retina is placed at the Image Focal Plane (IFP) of a thin lens (simplified model) so that the object can be seen at infinity [53]. Consequently, the retina can record an angular image by refocusing parallel rays. Indeed, the image is not modified by displacing the particle because the same scattered rays are always seen under the same angles (see Figure 3). We will show that this optical pre-treatment of the scattering signature dramatically simplifies the analysis of the image taken by the retina, removing the need for energy-costly image processing.

Note that such a setup can achieve position insensitivity if we consider a thin lens with infinite lateral extension. For a real lens with a finite pupil, the scattering signature is contained within the projection of the lens’ pupil on the retina. However, one must keep in mind that the projection of the lens’ pupil depends on the particle’s position, which will be discussed later.

### 2.4. Assymetric Crown Assembly and Signature Reconstruction

Let us consider a lens system coupled to a detector array. A major factor that limits the FoV is the pupil of the system. Our idea behind improving the FoV for light scattering analysis is to merge several of these subsystems around a perpendicular air channel in a crown shape, as illustrated in Figure 4a.

The edges of each subsystem may contain aberrations or unwanted artefacts that we call ‘blind fields’. By taking advantage of the symmetry of the scattering signature around the optical axis, one can sacrifice a bit of redundancy by facing half of the lenses toward the blind fields of the other halves. The resulting design is an asymmetric crown shape with an uneven number of optical subsystems arranged around a perpendicular air channel. An example is shown in Figure 4, with three subsystems (modelled as thin lenses).

In the IFP of each subsystem, one can reconstruct an experimental scattering signature $S_{exp}\left(\theta \right)$ (because there is an injunctive law linking a position on IFP and the scattering angle) along each associated sub-FoV. Figure 4c schematizes the signature seen through all three subsystems (FS, S, and BS, which stand for For-Side, Side, and Back-Side). As explained earlier, we design the sub-FoVs with an overlapping region (grey areas), in order to mitigate the defects at the edge of each sub-FoV by taking advantage of the asymmetric design.

As explained in the introduction, we try to retrieve certain particle’s parameters, such as diameter $D_{p}$ and complex refractive index n. To do so, we have to compare an experimental signature $S_{exp}\left(\theta \right)$ with a prediction from a model, usually based on the Lorenz–Mie, and, finally, perform the inverse problem. The optimization schematic used is presented in Figure 5.

A way to compare data retrieved from an experimental image with data from a modelled one has to be defined. For the lens-less setup, we’ve developed a procedure to reduce an image (either experimental or modelled) in order to compare those. This procedure requires a prior computation of the modelled image. All these steps, which are relatively expensive in calculation, had to be called regularly in an optimization loop. Moreover, as the image obtained depends on geometrical parameters such as the position of the particle $y_{p}$, these must be taken into account in the optimization (see, reference [31]).

In the Fourier-domain setup, the procedure is simplified. It requires performing the simple step of signature reconstruction $S_{exp}\left(\theta \right)$, and this only once. The computed signature $S\left(\theta ,\;D_{p},n\right)$ is retrieved directly from the Lorenz–Mie theory, and it is the only step within the optimization loop. The minimization criterion $\epsilon \left(D_{p},n\right)$ is also much simpler than previously, and is a simple mean of a one-dimensional deviation function:$$\epsilon \left(D_{p},n\right)\propto \int_{\theta \in FoV}^{}\left|S_{exp}\left(\theta \right)-S\left(\theta ,D_{p},n\right)\right|d\theta$$

In the future, one could test different definitions for $\epsilon \left(D_{p},n\right)$, using, for instance, an Euclidean norm instead of a simple deviation. It could also be interesting to implement ponderations at the center of each sub-FoV where the aberrations are softer.

Apart from the simplified image processing, such a sampling geometry allows for collecting the most of the scattered flux, especially around low scattering and backscattering angles where the scattering efficiency is usually high. This increases the sensitivity of the sensor to small particles compared to a lens-less setup.

Note that the illumination path must be completely free of optical elements, so that the sensitivity to scattering by diopters and ghost reflections is mitigated. This part is critical in order to ensure a good signal to offset ratio, as it was highlighted in a previous version of this design, with five subsystems (see, reference [54]).

### 2.5. Compact Optical Subsystems with Coincidental Fourier Planes

Each optical subsystem must meet a certain set of specifications: it must be as compact as possible; it must also fold its Fourier plane (i.e., IFP) onto a plane perpendicular to the fluidic channel so that all subsystems’ Fourier planes are coincidental. This set-up enables the use of a single holed retina to image all Fourier planes on separate regions of interest (ROIs).

As in the previous example, we select an assembly made with three subsystems: a first 90° centered system called S (‘Side’) subsystem and two symmetrical 90 ±30° called FS and BS (‘Front-Side’ & ‘Back-Side’). Those subsystems are composed with a front spherical diopter, an internal reflector and a flat back diopter. The reflecting surface can be coated with a metallic reflective layer such as gold. In this section, the design and analysis tools are made using ray tracing (RT) tools (Zemax Optic Studio, Sequential mode). Let us focus first on the S subsystem illustrated in Figure 6a (note that the (X,Y) coordinate system is used only in this subsection, and is specific to a subsystem alone. A new coordinate system will be used later when each subsystem will be arranged around the air channel).

The system is designed with full compatibility with a sub-retina and has a reasonably long Back Focal Length (BFL = 1 mm). This allows us to have an efficient cloaking device with efficient stray light protection that fits within the BFL (between the optical piece and the retina). Only the front surface is curved, the back surface being a simple flat surface unprocessed from the substrate, thus removing the need for a backside process. The front diopter is slanted by 10° in order to facilitate an eventual unmolding process. Then, we can improve the uniformity of the PSF (Point Spread Function) by engineering an internal reflective surface that corrects aberrations. This reflector is a freeform surface defined by a Chebychev 2D-polynomial [55]. Due to the relative position of the front surface and the sub-retina, the reflector is slanted at 33.5° so that the optical axis is rightfully aimed at the center of the sub-retina. The sub-FoV is as wide as 60°.

The subsystem takes advantage of three surfaces out of a same volume, thus can be manufactured out of a unitary piece. For this prototype, the material used for the design is fused silica glass, but it can also operate with most optical polymers. Compactness is then improved by truncating such optical surfaces, as seen in Figure 6.

For the FS (or BS) subsystem (Figure 6b), the same rules were applied. However, the image plane (half a sub-retina, the second sub-retina being shared by the FS and BS subsystems) is rotated by 30° with respect to the front surface because a different viewing angle of scattering signatures is evaluated.

In Appendix A, we plot the modulation transfer function of a subsystem, which is very useful to quantify the resolution of an optical system; it may however be difficult for a non-specialist to visualize the effects of the optical system on an image. Thus, we perform image simulations from an object at infinity (angular object) by the three subsystems (see, Figure 7).

The object is a square angular example picture sized 60° × 60° (see, Figure 7a). We compute the spatially variant PSF of the optical subsystem. Chromatic aberrations are also taken into account as the PSF is calculated with RGB colors, this information can be useful if one wants to scatter light with different wavelengths. The simulation image is finally obtained by convoluting the object and the PSF.

Figure 7b shows the image simulation obtained with the S optical subsystem and Figure 7c with both the FS and BS subsystems. As expected, the FS and BS images are shared from a first sub-retina, and rotated by ±30°, as explained earlier. The magnification is rightfully adapted to our holed retina and the PSF appears good enough for scattering signature imaging.

Figure 7d presents a top-down view of the geometry, showing the relative position and tilt of the subsystems with respect to the sub-retinae. For readability reasons, only the front surfaces, internal reflectors and retinae are drawn. The FS and BS subsystems are symmetrical with respect to the separation plane of the two halves of the second sub-retina.

### 2.6. Merging the Subsystems and Coupling with the Holed Retina

All three optical subsystems (S, FS and BS) are merged into a unitary piece. Front surfaces are at an equal distance (Front Focal Length, FFL = 0.5 mm) from the center of the channel. The channel is designed to have a section similar to that of the holed retina. The resulting 3D model is rendered in Figure 8. It contains the merged subsystems around a vertical fluidic channel. We also show, in the figure, the arrangement of the optical piece in relation to the holed retina.

We have designed a cloaker optimized for stray-light management. In particular, it is built with stray light protection features than can fit within the BFL of the optical piece. Such a cloaker, which is presented in Figure 9, is printed with a SLA (Stereo-Lithography Apparatus) 3D-printer (Form3 by Formlabs) using adaptive voxel resolution (down to 25 µm) and made with a black photopolymer resin. It is sized to fit within the socket of the Creapyx motherboard. It contains housing for the optical piece and shows optimized apertures in front of all three subsystems (S, FS and BS subsystems).

Note that the illumination system is not directly integrated with the optical setup. This choice is motivated by the fact that such an optical piece can be studied separately from the illumination system, and that consequently, the latter can be interchanged easily, within the framework of an iterative approach to development. Moreover, separating the illumination system from the optical piece allows for implementing elements that can protect the detectors from stray light coming from the source.

Thus, we chose to implement a 45° mirror (3 mm wide), which sits on the cloaker and is separated from the optical piece by a diaphragm printed with the cloaker (see Figure 9a). By doing so, one can insert an interchangeable illumination module, as long as it provides a downward beam that recovers the pupil of the mirror. Preferably, such an illumination system would be designed in a similar way to the miniature refractive-refractive optics present in the optical piece itself, and would allow forming a beam over the fluidic channel from a bare laser diode (LD). In our case, for reasons of supply, we use a system consisting of a commercially available bare LD (637 nm, 5 mW), and aspherical lenses contained in a standard 0.5 inch optical tube. Further details on the illumination module will be provided in Appendix B.

Below the optical piece, the shell has optimized apertures that can select the scattered rays that were transformed by the optical system, as shown in Figure 9b. These apertures are slanted with the angle defined by the folding mirror.

### 2.7. Angular Response at the Coincidental Fourier Plane

We will now evaluate the angular response of the merged system. In Figure 10, we present a RT simulation (Zemax Optic Studio, non-sequential mode, the non-sequential mode is the Monte–Carlo ray-tracing method where rays can interact with any surface of an arbitrary geometry) of the silica glass piece associated with the cloaker and its optimized apertures. For a better comprehension, we have simplified the scattering particle down to a simple conical source point, which has a given irradiance angle, and is axially symmetrical around the beam axis. An image monitor is placed at the coincidental Fourier plane, and can record the irradiance pattern induced by the scatterer.

By sweeping the angle of the conical source from 1° to 180°, one can obtain the intensity map on the coincidental Fourier plane for each scattering angle and then, associate a scattering angle (and intensity) to each pixel of the monitor. In Figure 11a, we plot the angular response of the optical system that evaluates a centered scatterer S(0,0,0). The angle map is plotted using a colored scale, and the associated intensity is plotted using a transparent-black (bright-dark) overlay.

We have drawn white rectangles that correspond to the two sub-retinae. We recognize the three regions of interest (ROIs) associated with the three optical subsystems (FS, S, BS). We verify that a point within a ROI is associated to a single scattering angle. These ROIs allow us to evaluate the scattering signature thanks to the associated overlapping FoVs. Again, we verify that the S-ROI falls on a first sub-retina and that the FS and BS ROIs share the second sub-retina. The total FoV is as wide as 110°.

In order to evaluate the impact of the position of the scatterer, we perform the same RT simulation but the scatterer is displaced at $S\left(x_{p},y_{p},z_{p}\right)$, using reasonable positions when compared with the incident beam ($x_{p}=100\;\mu m,\;y_{p}=15\;\mu m,\;z_{p}=10\;\mu m$). Note that the range of accessible positions is highly elongated along the axis of the illumination beam (*X*-axis). The result of such a simulation is presented in Figure 11b. We verify that every pixel is still associated with the same scattering angle when compared to the centered scatterer case (pixels are colored the same in both subfigures). This gives us confidence that our strategy of measuring a scattering signature with position insensitivity is relevant.

The main difference between the two images is that the images of the pupils of each subsystem are slightly transformed due to the displacement of the scatterer, which was to be expected. This should not present many difficulties as it takes effect at the edge of every subsystem (blind fields), which can be easily corrected thanks to our asymmetric assembly of subsystems

## 3. Fabrication Process

### 3.1. Micro-Machining on Glass

The prototype of the monolithic optical piece is a challenging piece to manufacture. It has small dimensions and the surfaces of the diopters must have optical-grade surface qualities. Precision micro-tooling of molds [56] is still very expensive but can be cost-efficient if optical pieces are molded at volume fabrication (using for instance micro-injection molding [57]). Direct laser 3D printing by two-photon polymerization shows great promise printing miniature optics [58,59] but is not yet suited for printing millimeter-sized optical pieces: the lateral dimensions are limited to a few hundred micrometers.

For our prototype, we have selected a manufacturing process based on an innovative laser micro-machining of glass [60,61] developed by FEMTOprint that relies upon three main steps: laser exposure, wet/chemical etching and polishing.

The outline of the 3D shape of the device inside a bulk piece of fused silica is generated by sweeping a tightly focused laser beam at a wavelength of 1030 nm. The exposure to femtosecond laser pulses triggers a non-linear absorption process that causes a glass densification within the laser voxel (the laser voxel is the volume at the vicinity of the focusing point where the optical power density has reached a certain value, allowing for the non-linear densification effect) that eventually induces a drastic change in selectivity (up to 1000 times) when immersed in an etching solution [62]. The excess material is then removed by a KOH-based solution and the unexposed volume of the device is released from the wafer.

Machined surfaces show a typical roughness *Ra* (Ra stands for Roughness average, and is the average value of the profile heigh deviations) larger than approx. 100 nm, i.e., not low enough for optical elements such as the channel-facing front diopters, the reflecting surfaces, and the back diopters. For this reason, a proprietary polishing process has been applied to selectively reduce the roughness of the optical elements by locally reflowing the material surfaces while minimizing the deformation. To this goal, the slanted freeform surfaces have been designed to be easily accessible to the polishing tool and a reduction of *Ra* to values between 5 and 15 nm has been achieved, which is much better that the roughness initially targeted (λ/10), and allows for minimal surface scattering. There was no need to polish the flat top and bottom surfaces, as these were not machined and therefore maintained a pristine surface quality.

An optical image showing a top view of the fabricated glass piece is shown in Figure 12.

Figure 12c shows an optical view of the polished glass piece. Note that the rough surfaces (machined surfaces without polishing) appear bright because they scatter light, whereas the polished and pristine surfaces (e.g., the slanted freeform surfaces) appear dark, which testifies to an excellent surface quality. As a comparison, Figure 12d is a top-down view of the original drawing, which shows the great fidelity of the fabricated piece compared with the model.

Note that a molding and replication process was explored for a previous design of this optical piece in reference [54]. The use of molded polymer pieces would allow formanufacturing the sensor inexpensively, at volume fabrication.

### 3.2. Mirror Deposition

In this section, we describe how the mirror deposition step is performed. A metallic layer has to coat at least the slanted freeform surfaces, while avoiding the first diopters facing the fluidic channel. Thus, a specific area must be covered prior to any metal deposition step: a stripe above the fluidic channel. The remaining surfaces that do not have optical functionality can be metallized without affecting the operation of the system. We cover those surfaces using a stencil made from a 1-mm-wide stripe of Kapton (polyimide adhesive film) as seen in Figure 13a. In a preliminary process, we cut the said Kapton stripe manually with a blade and a pair of binoculars. We position, again manually, the stripe on the optical piece using a pair of tweezers. In a future process, the fabrication of such a stencil should be computer-assisted; for instance, via CNC cutting, laser ablation or lithography.

The metallic layer is deposed by physical vapor deposition (PVD): first with a 10 nm titanium adhesion layer, followed by a 100 nm gold layer. Preliminary studies have shown that the silica/metal interface is well preserved, in terms of surface quality, which justifies the use of such process to manufacture internal reflectors. The use of gold as the coating material is motivated by its reflectivity on a range from red to NIR (>90%), which is weakly impacted by the titanium layer. After this step, the parts that were not to be metallized are revealed, by simply removing the Kapton stencil with tweezers (see Figure 13b). The final optical piece, with its selectively deposited mirrors is presented in Figure 13c.

In the future, a specific mirror deposition process will have to be developed in order to be compatible with our polymer pieces.

## 4. Characterization

### 4.1. Experimental Setup

The characterizations of the sensor are performed in the same fashion as for the lens-less setup [31], by using the Creapyx motherboard that drives the holed retina, and a calibrated aerosol test bench (Constant output atomizer model 3076 by TSI) that provides a continuous flow of calibrated particles, which are commercially available monodisperse PSL beads in our case.

A photography of the characterization setup is given in Figure 14a. The optical piece is sitting on the cloaker that properly aligns it with the holed retina below. As explained earlier, the downward illumination is provided by the illumination module (standard 0.5 inch lens tube, with a LD and aspheric lenses, see Appendix B), and redirected horizontally toward the muzzle of the air channel using a 45° mirror.

In the photography we see that, despite the diaphragm, a large portion of stray light comes toward the optical piece, hitting some of its diopters. Then, we take a reference image, using the holed retina when no aerosol is injected in the channel. Such a reference is presented in Figure 14b, and is automatically subtracted from future images. We notice that a portion of the second sub-retina is impacted by stray light (FS and BS ROIs), but not to a great extent.

Given the visible amount of stray light hitting the optical piece, and the reasonable amount that ends up on the retina, one might argue that the anti-stray-light designs were successful, at least for our given illumination setup.

We record bursts of 32 images, in global shutter (GS) mode, and we follow the standard deviation of the full image (with its reference subtracted). Then, an image is saved when its standard deviation reaches a given threshold. More information on the experimental setup, notably in terms of the sampled aerosol and the driving of the retina, is reported in reference [31].

### 4.2. Experimental Images of Scattering Signatures

We were able to detect single PSL particles of 3 µm and 830 nm. Examples of representative signatures that can be experimentally recorded are given in Figure 15a,b. The images obtained seams quite repeatable for a given aerosol, both in terms of brightness and patterns. A precise study on repeatability should be conducted in the future (in order to quantify the deviation), but this can already give us confidence that the signature is weakly impacted by particle position, due to the Fourier-domain imaging setup.

We observe three separated areas that correspond to the S-ROI on the first sub-retina (rows 76 to 150); the BS and FS-ROIs on the second sub-retina (rows 1 to 75). We notice that the positions of such ROIs are slightly translated compared to what was expected, due to the small alignment problems that will have to be addressed in the future. The white pixels correspond to pixels from the reference image that reached a certain threshold, that were removed from the image processing, in order to be less sensitive to Poissonian-type noises.

The scattering signatures of large spheres, such as the 3 µm PSL, exhibit a number of Mie’s oscillations, whereas the 830 nm PSL does not (at least within the evaluated FoV). This effect is predicted by the Lorenz–Mie theory (spheres with refractive index $n_{PSL}=1.5875$ [53]), and be can be found on the RT simulations for the associated PSL diameters (see, subfigures (c,d)).

Note that the images were normalized, but the same color axis was kept between (a,b) on the one hand and (c,d) on the other hand, in order to compare the relative brightness of scattering signatures obtained with different diameters.

### 4.3. Reconstruction of Scattering Signatures

The oscillations of 3 µm PSL signatures facilitate the interpretation of such a recorded signature. For didactic reasons, we are going to focus in particular on the specific image shown in Figure 15a. We observe several vertical oscillations on the S-ROI, whereas those on the BS and FS ROIs are slanted by ±30°, which was expected by design. Those oscillations follow what we call the iso-θ, which are defined as the trajectories on the retina that correspond to the rays scattered with the same scattering angle θ.

The procedure to reconstruct a signature is relatively simple: let us first consider one of the three ROI separately. First, every pixel in the ROI is normalized using the simulated intensity map calculated previously using a RT simulation (intensity map in Figure 11a). Then, by using an angle map, also computed by RT (see, Figure 11a), one can associate each pixel of the retina to a scattering angle. For a given scattering angle (with a tolerance of 1°), a mean value of the intensity carried by those associated pixels can be calculated which then results in a sub-scattering signature reconstructed within the sub-FoV of each ROIs. In this case, the ROIs are misaligned with respect to the retina, thus, we had to manually translate the angle map to best fit with the recorded image. Due to this effect, the BS ROI was cropped, resulting in a loss of FoV for very high angles. Still, the total FoV is relatively wide, at about 85°.

Before plotting those sub-signatures together, we had to perform a correction that consists in having a multiplicative factor for each sub-signature. This step is motivated by the fact that the relative brightness of the ROIs are not consistent to what was expected: this can be seen especially with the BS-ROI (see Figure 15a) that is much brighter than expected (see Figure 15c). Hopefully, this correction appears to be very repeatable between the different recorded images, and could be easily found through a calibration campaign. We believe that it could be attributed to misalignment errors of the apertures of the cloaker with respect to the subsystems’ pupils, which may create an uneven loss of photometry between the ROIs.

One notices that the sub-signatures are not perfectly overlapping: the edges of the individual curves correspond to the blind field and should not be over-trusted. Again, this overlapping mismatch will not overly affect results as it appears to be very repeatable, and thus easy to correct in post-process; for example, with a single calibration. To connect the curves, we make averages on the overlapping regions. The averaging weight varies linearly from 1 to 0, dropping at the edge of each curve. With this step, one can have a continuous merged curve (drawn in black), which is more representative of real scattering signatures.

We insist on the fact that the reconstruction step is particularly resource-efficient in terms of computing capabilities, especially when applying only matrix operations (which are optimized by MATLAB, our data processing tool), at least compared to the planar projection set-up (see, reference [31]). This gives us confidence that procedure is compatible with low-energy portable applications.

In Figure 16, we plot the merged scattering signatures alongside the theoretical signature of a 3 µm PSL sphere (refractive index $n_{PSL}=1.5875$ [53]) illuminated by a 637 nm wavelength beam, computed using the Lorenz–Mie theory.

Based on such a representative plot, we find that the reconstructed signature is surprisingly faithful to the theoretical one, especially with the position and frequency of the Mie’s oscillations in the plot (which were not corrected).

This gives us confidence that such an architecture of the sensor and reconstruction procedure is well suited for our application. However, to conclude on this point, we will have to conduct a proper statistical study, over a wide number of PSL diameters, with blind reconstructions. To do so, we will have to make our setup more robust. In the near future, the 3D printing of the cloaker will be optimized; then, the next step would be to integrate directly both the cloaker (with its optimized apertures) and the optical piece on-top the CMOS at wafer scale, in order to achieve optimal alignments.

Note that a study on a larger number of particles, and on a larger diameter range, could allow us to conclude on the detection limit in terms of particle size. We can already determine with which SNR (Signal to Noise Ratio) we have detected 3 µm and 830 nm PSL. The PSL of 3 µm were detected with an SNR of about 825, while the PSL of 830 nm (which scatter less intensely) were detected with an SNR of about 145 (these values were quite constant between images obtained from the same PSL diameter, meaning that the brightness and patterns of the recorded signatures are very repeatable).

Those values are quite high already, and thus we can realistically expect that we can measure smaller particle sizes using such a setup. In addition, the SNR should be further improved by optimizing the illumination system so that the amount of stray light is minimized.

## 5. Discussion about Particle Identification

We recall that the principle of measuring a scattering signature is to find certain parameters of the particles that compose an aerosol, such as diameter $D_{p}$ and refractive index n. We compare the image recorded experimentally, with a prediction from a model (Lorenz–Mie theory), and a geometrical transformation in order to perform the inverse problem. To do so, we define the minimization criterion $\;\epsilon \left(D_{p},n\right)\propto \int_{\theta \in FoV}^{}\left|S_{exp}\left(\theta \right)-S\left(\theta ,D_{p},n\right)\right|d\theta$, which evaluates the deviation between an experimental signature $S_{exp}\left(\theta \right)$ and a modelled one $S\left(\theta ,D_{p},n\right)$. Here, we only study the case of a non-absorbing particle, thus we consider the refractive index as a real quantity. In the case of an absorbing particle (such as a carbon particle), the imaginary part Im(n) must be added to the list of parameters to be optimized, along with $D_{p}$ and Re(n).

The optimization procedure, which was shown in Figure 5, is used. Here, we are going to study its accuracy, in its capacity to find the correct values of $D_{p}$ and n. For didactic reasons, we are only considering the case of the 3 µm PSL. The test images are found in Figure 15a. The minimization criterion is plotted in Figure 17 as a function of $D_{p}$ and n.

We draw the targeted parameters as black lines, and the reasonable local minima as a white contour. The accuracy of the procedure is judged by the extent of such a contour, and its distance from the targeted parameters. In addition, we are also interested in local minima, i.e., their number and prominence, for the reason that they can induce important errors in the estimation of the desired parameters.

We have already discussed that the procedure used in the lens-less setup was less than optimal in view of these different judgment criteria (see reference [31]). In comparison, the procedure for the Fourier-domain setup appears more successful, at least for the test case of a 3 µm PSL. We judge, however, that such a procedure is to be improved, not least because the precision is still insufficient, in particular by the fact that the valley of the local minimum is very elongated in one direction, and by the presence of another local minimum. The direction of elongation of the local minima seems to come from the fact that similar signatures can be obtained for different couples $\left(D_{p},n\right)$, and that a simple measurement of the scattering signature only allows to find a kind of optical diameter (that combines both $D_{p}$ and n). We believe that it is possible to solve this limitation by simultaneously evaluating the optical diameter of the same particle (of fixed geometric diameter $D_{p}$) in two different ways, for example by recording a dichromatic scattering signature.

If one knows the type of particle evaluated and its refractive index, the accuracy on the geometric diameter is given by the horizontal width of the minimum in the plot, which results in an error of about 10% of the value obtained. It is likely that the accuracy is dependent on the diameter. In future work, it will be necessary to perform this accuracy study for different particle diameters. In particular, this will be necessary for small particles, such as those of 830 nm, in order to know the result of the method when no Mie oscillation is found in the signature.

Then, another way to improve this procedure is to make the assembly more robust, as explained previously. Other elements could also help to reduce the uncertainty in the retrieval for parameters $D_{p}$ and n, such as, for example, using the overall image brightness as a starting point for the optimization.

Again, the simple use of a 3 µm PSL example is far from sufficient to conclude on the accuracy of the procedure. To do so, we will have to perform a proper statistical study on a large number of particles, and over a wide range of diameters and types of particles (organic, inorganic, aqueous, and irregular).

## 6. Conclusions

In conclusion, we have designed a PM sensor that features a miniature, monolithic refracto-reflective system suited for the optical pre-treatment of scattering signatures. This design is applied to our family of PM sensor prototypes that uses holed retinae. It addresses the various areas of improvements for miniature PM sensors that were identified in a previous lens-less setup [31], such as the ability to collect more flux from the scatterer (with viewing angles where scattering is usually strong), particle’s position insensitivity, ultra-wide field of view, and simplified image processing. It features a monolithic assembly of three compact optical subsystems, arranged around an air channel, that are folded onto a coincidental focal plane. The optical piece includes a free-form surface and shows greater performances in all fields. In particular, it features elements that prevent stray-light sensitivity and is fully compatible with the holed retina: a custom fluidic-optronic CMOS chip.

We present a state-of-the-art fabrication process for a glass prototype that was developed by FEMTOprint, which consists of the direct, 3D writing on glass, using a femtosecond laser inscription and selective etching and local surface polishing, followed by a selective mirror deposition process.

The glass piece was assembled with an optimized cloaker and the holed retina, and tested with calibrated aerosols. We were able to measure the scattering signature (optically processed) of single particles of polystyrene beads of different sizes. We were then able to test the reconstruction procedure of the scattering signature, which is particularly faithful to a theoretical signature. This gave promising results in solving the inverse problem, which consists of finding the size and refractive index of a particle using the Lorenz–Mie theory.

However, in order to conclude on the performance of such a system, we will have to carry out more in-depth tests (on an assembly that needs to be made more robust). For example, it would be interesting to record signatures from a larger quantity of particles in order to perform a statistical study. Such a study would allow us, for example, to relate the detection rate to a particle concentration.

We would also test our device on calibrated particles of various sizes, which would allow us to experimentally estimate a Limit of Detection (LoD) in size. Then, in order to be more representative of a real aerosol, we would perform those tests on a larger variety of particle types (such as organic, inorganic, aqueous, irregular particles etc.).

## 7. Patents

Some of the designs reported in this article are filed under the following patents:Boutami, S. and Nicoletti, S., Optical detector of particles, 2018, Patent Application EP3574301, WO2018138223, FR3062209, KR20190112049Jobert, G., Optical particle detector, 2018, Patent Application EP3598102, US20200033246, FR3083864.

## Figures and Tables

**Figure 1 sensors-21-03181-f001:**
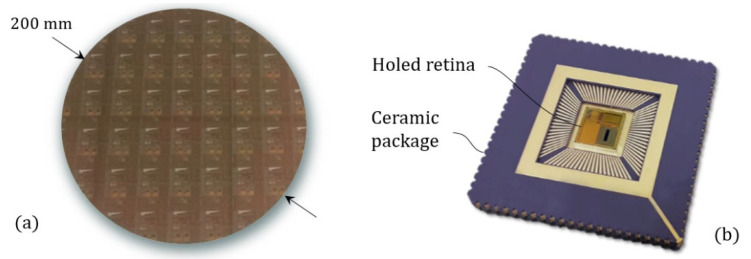
(**a**) Photography of a 200 mm wafer that contains the custom image sensors. (**b**) Photography of a holed retina bonded on the ceramic carrier.

**Figure 2 sensors-21-03181-f002:**
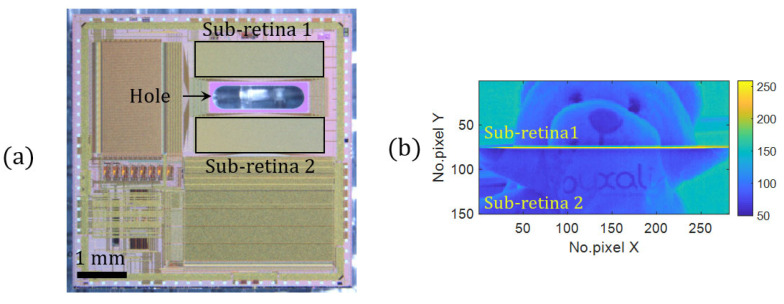
(**a**) Optical image of the fabricated holed retina. (**b**) Image example obtained focus mode.

**Figure 3 sensors-21-03181-f003:**
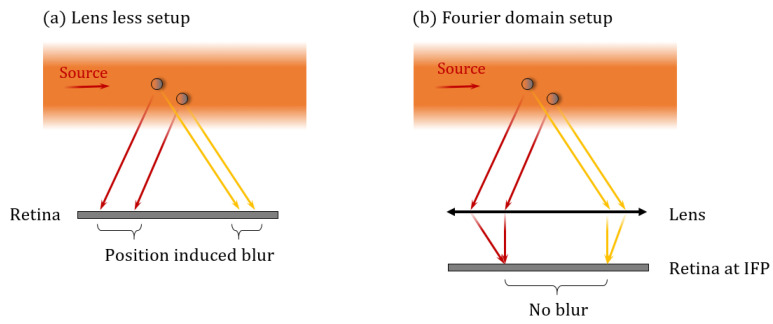
Illustration of the d-f setup that performs angular imaging by refocusing parallel scattered rays.

**Figure 4 sensors-21-03181-f004:**
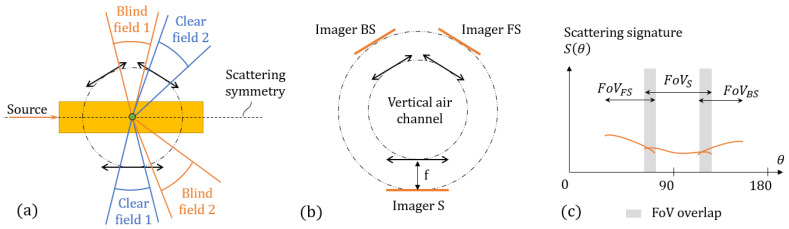
(**a**) Principle of blind fields asymmetrically coupled to clear fields. (**b**) Lenses coupled with imagers in Fourier planes. (**c**) Principle of overlapping sub-FoV for signature reconstruction.

**Figure 5 sensors-21-03181-f005:**
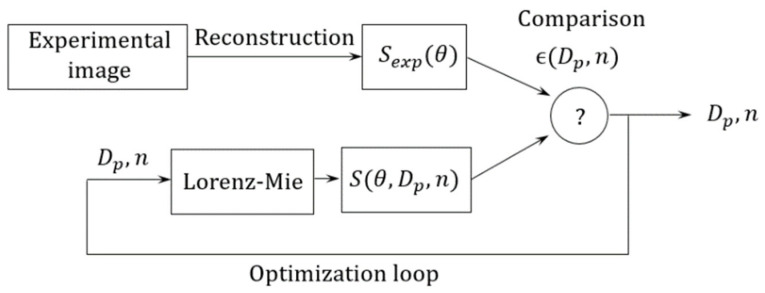
Schematic of the optimization procedure used for particle characterization.

**Figure 6 sensors-21-03181-f006:**
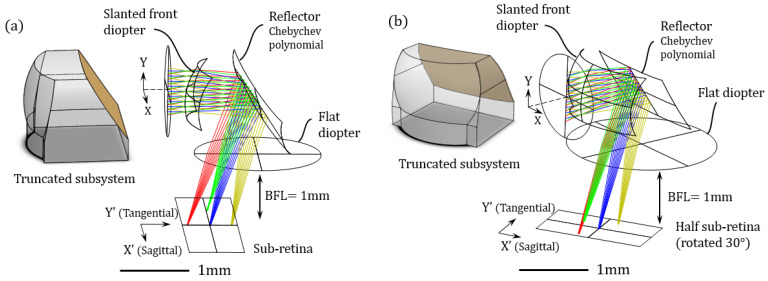
(**a**) Design of the S subsystem alongside its truncated model. (**b**) Design of the FS/BS subsystem alongside its truncated model.

**Figure 7 sensors-21-03181-f007:**
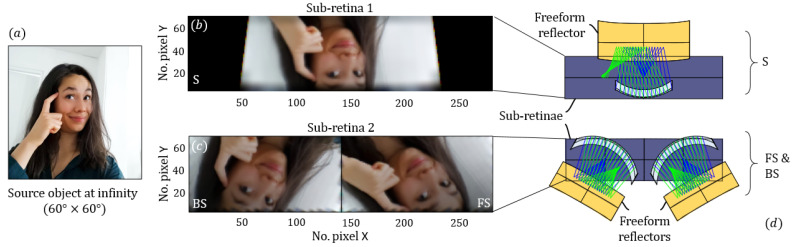
(**a**) Example picture as a source object at infinity. (**b**) Simulated images obtained with the S subsystem on the first sub-retina and (**c**) with the FS and BS subsystems sharing the second sub-retina. (**d**) Schematic top-view of the three subsystems aligned with the two sub-retinae.

**Figure 8 sensors-21-03181-f008:**
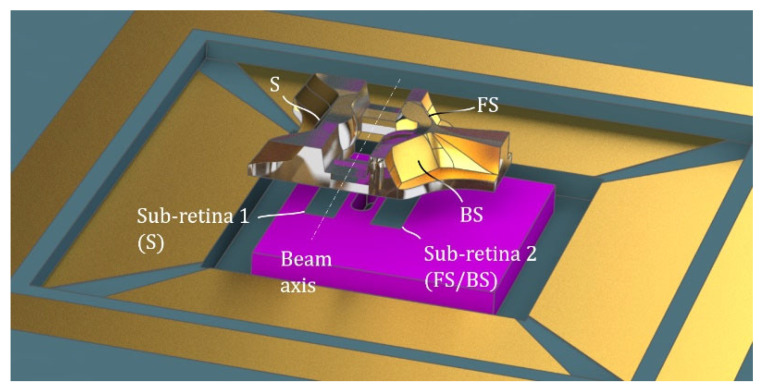
3D view of the monolithic optical system aligned with the holed retina.

**Figure 9 sensors-21-03181-f009:**
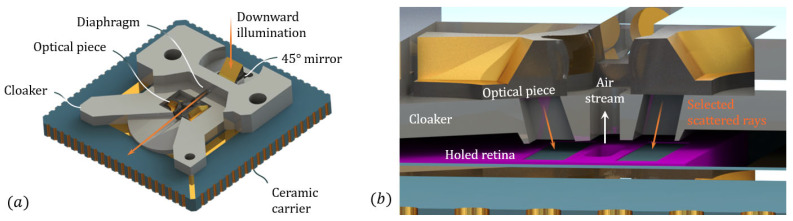
(**a**) 3D view of the optical piece with the cloaker and the holed retina mounted on ceramic. (**b**) Cross-sectional view of the optical setup, showing the optimized apertures.

**Figure 10 sensors-21-03181-f010:**
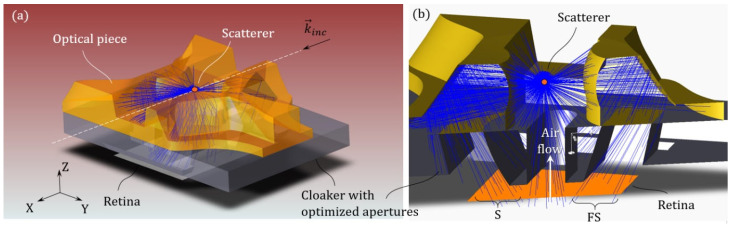
(**a**) 3D view of the RT simulation. (**b**) Close-up cross-sectional view.

**Figure 11 sensors-21-03181-f011:**
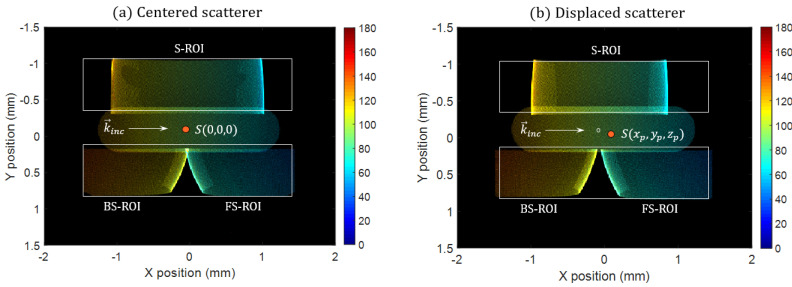
Angular response (in degrees) of the optical system evaluating (**a**) a centered scatterer and (**b**) a displaced scatterer.

**Figure 12 sensors-21-03181-f012:**
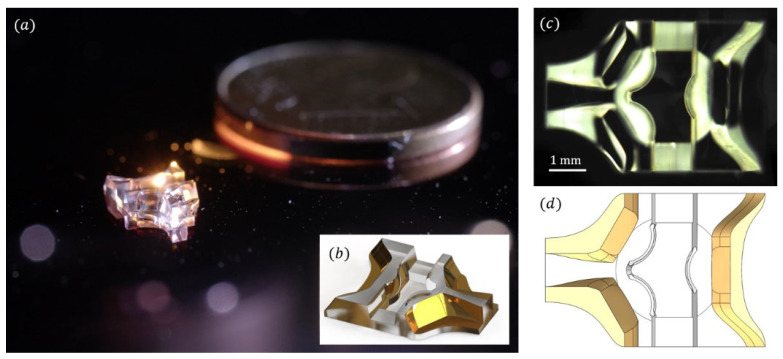
(**a**) Photography of the fabricated optical system next to a one-euro cent coin. (**b**) 3D view of the optical system. (**c**) Optical view of the fabricated glass piece. (**d**) Associated view of the original drawing.

**Figure 13 sensors-21-03181-f013:**
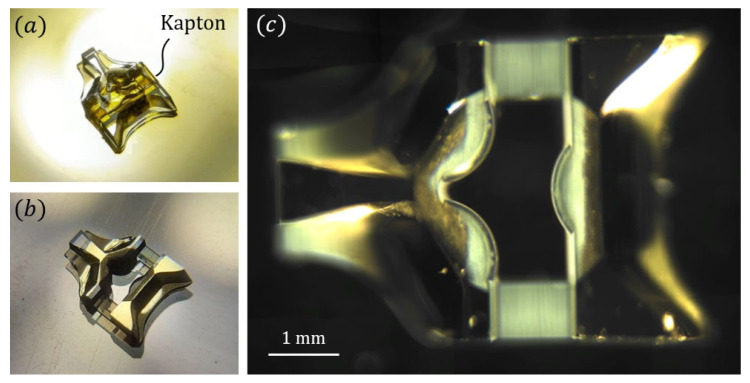
(**a**) Photography of the optical piece with the Kapton stencil. (**b**) Optical piece after the metallization process and stencil removal. (**c**) Optical top-view of the final optical piece.

**Figure 14 sensors-21-03181-f014:**
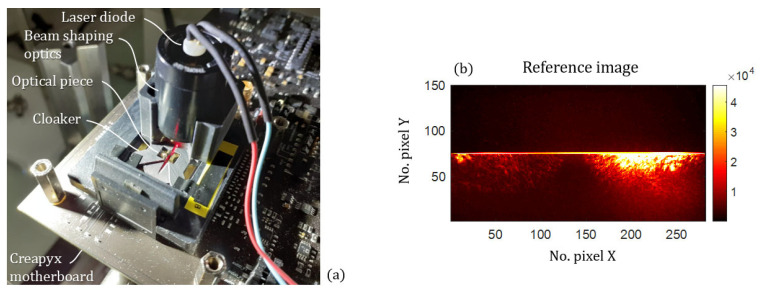
(**a**) Photography of the characterization setup. (**b**) Reference image without particle.

**Figure 15 sensors-21-03181-f015:**
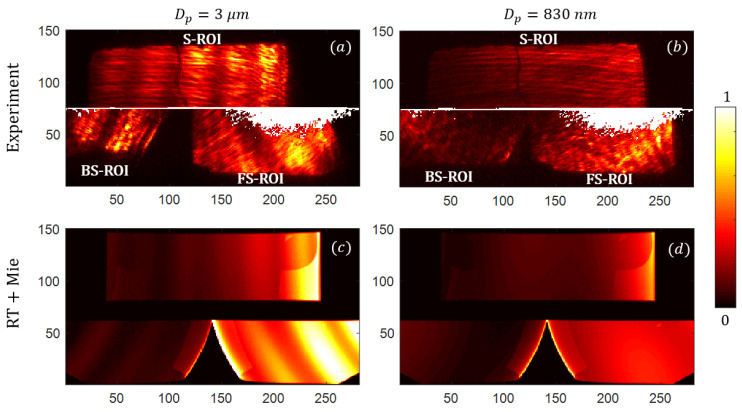
Experimental images of scattering signatures of PSL spheres: (**a**) 3 µm and (**b**) 830 nm; compared with associated computed signatures: (**c**,**d**) respectively.

**Figure 16 sensors-21-03181-f016:**
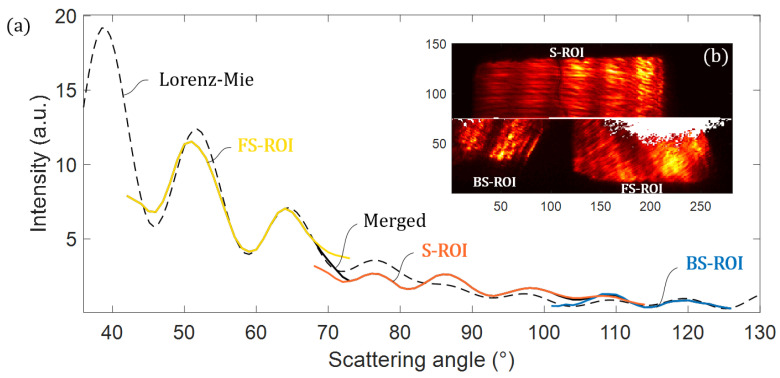
(**a**) Reconstructed signature from experimental image plotted with the theoretical signature. (**b**) Experimental image of a 3 µm PSL used for the reconstruction.

**Figure 17 sensors-21-03181-f017:**
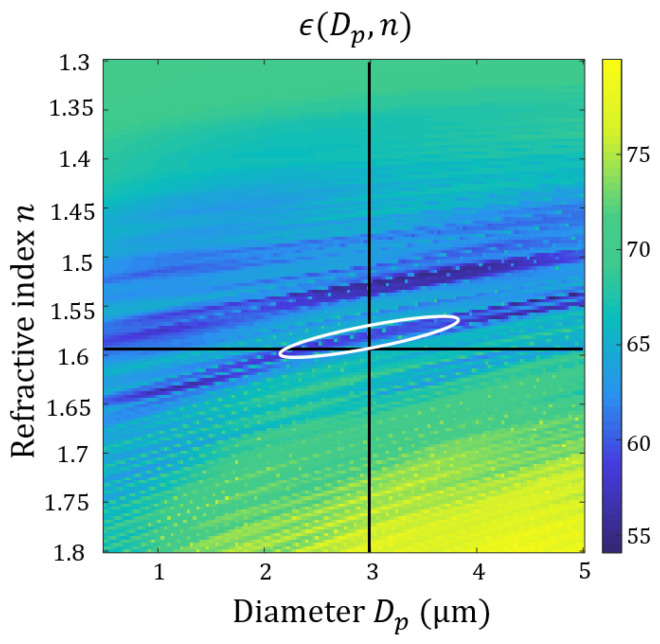
Map of our minimization criterion versus particle diameter and refractive index in logarithmic scale.

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
