# Peer review of "Miniature Optical Particle Counter and Analyzer Involving a Fluidic-Optronic CMOS Chip Coupled with a Millimeter-Sized Glass Optical System†"

_sensors, 2021, doi:10.3390/s21093181_

Round 1

Reviewer 1 Report

Overall, I think this is a novel low-cost PM sensor design. BY using CMOS sensor, the device can realize PM sizing and identification. However, the structure of the manuscript needs to be reorganized. Two major shortcomings are very distractive. First, the resolution of figures is too low. Second, too many details regarding sensor design distract readers. Detailed comments are listed as below:

  1. Please provide the full name of “CMOS”
  2. I highly recommend the author reconstruct the introduction part. For example, Figures 1 and 2 are not necessary. The author can include references containing similar figures.
  3. Not sure whether it is due to low resolution, Figure 4(b) does not seem very persuasive. I would recommend changing Figure 4(c).
  4. Too many details regarding sensor design are included in the main text. I would suggest moving them to supplementary materials. Readers may be more interested in the performance of sensors. For example, Figures 17 is more attractive.
  5. Figure 19 should be in the method section instead of the results section.
  6. In the manuscript, the author did not discuss when a refractive index has both a real part and an imaginary part (refractive index = n+mi). Will this situation interfere with the method?
  7. What is the resolution of the proposed sensor design? The manuscript provides results for 830 nm and 3 um. Can it distinguish 830 nm and 800 nm? Or can it distinguish 830 nm and 1 um? This information is very useful to readers.

Reviewer 2 Report

This nice paper concerns the latest advance in the realization of a very miniaturized optical counter, mainly for the detection of PM2.5 and PM1 particles. The paper is mainly technical and presents in detail the instrument design and all the steps the authors have conducted to realize a prototype. The first results are very promising, although, as said honestly by the authors, a large amount of work is still needed to improve the quality of the measurements, the data processing, and the calibration for non-spherical particles.

The paper can be published after considering these few comments:

Line 253: Can you define “BFL”?

Line 270: Can you explain what means “non sequential mode”?

Line 478: Can you define what “PVD” is?

Line 524: Can you better quantify “are quite repeatable”, for example by giving a measured intensity dispersion or standard deviation?

Line 565: The red lines are difficult to see.

Round 2

Reviewer 1 Report

The authors have addressed most of my comments. The only concern is that the manuscript is a little bit lengthy. I would recommend for publication.